# A machine learning approach using partitioning around medoids clustering and random forest classification to model groups of farms in regard to production parameters and bulk tank milk antibody status of two major internal parasites in dairy cows

**Andreas W. Oehm** [1] *, **Andrea Springer** [2], **Daniela Jordan** [2], **Christina Strube** [2], **Gabriela Knubben-Schweizer** [1], **Katharina Charlotte Jensen** [3,4], **Yury Zablotski** [1]

**1** Clinic for Ruminants with Ambulatory and Herd Health Services, Ludwig-Maximilians-Universität Munich, Oberschleissheim, Germany, **2** Institute for Parasitology, Centre for Infection Medicine, University of Veterinary Medicine Hannover, Hannover, Germany, **3** Clinic for Cattle, University of Veterinary Medicine Hannover, Foundation, Hannover, Germany, **4** Institute for Veterinary Epidemiology and Biostatistics, Freie Universität Berlin, Berlin, Germany

* Andreas.Oehm@outlook.com

## Abstract

*Fasciola hepatica* and *Ostertagia ostertagi* are internal parasites of cattle compromising physiology, productivity, and well-being. Parasites are complex in their effect on hosts, sometimes making it difficult to identify clear directions of associations between infection and production parameters. Therefore, unsupervised approaches not assuming a structure reduce the risk of introducing bias to the analysis. They may provide insights which cannot be obtained with conventional, supervised methodology. An unsupervised, exploratory cluster analysis approach using the k–mode algorithm and partitioning around medoids detected two distinct clusters in a cross-sectional data set of milk yield, milk fat content, milk protein content as well as *F. hepatica* or *O. ostertagi* bulk tank milk antibody status from 606 dairy farms in three structurally different dairying regions in Germany. Parasite–positive farms grouped together with their respective production parameters to form separate clusters. A random forests algorithm characterised clusters with regard to external variables. Across all study regions, co–infections with *F. hepatica* or *O. ostertagi*, respectively, farming type, and pasture access appeared to be the most important factors discriminating clusters (i.e. farms). Furthermore, farm level lameness prevalence, herd size, BCS, stage of lactation, and somatic cell count were relevant criteria distinguishing clusters. This study is among the first to apply a cluster analysis approach in this context and potentially the first to implement a k–medoids algorithm and partitioning around medoids in the veterinary field. The results demonstrated that biologically relevant patterns of parasite status and milk parameters exist between farms positive for *F. hepatica* or *O. ostertagi*, respectively, and negative farms. Moreover, the machine learning approach confirmed results of previous work and shed

**Data Availability Statement:** All relevant data are included within the manuscript and its Supporting Information files. The data sets used to obtain the results presented in this manuscript are provided as supplementary files with farm id removed in order to ensure anonymity. Furthermore, the questionnaires to collect information on farming type and pasture access are included both the German original as well as an English version.

**Funding:** Farm visits and data collection in the context of the underlying cross-sectional study were financially supported by the German Federal Ministry of Food and Agriculture (BMEL) through the Federal Office for Agriculture and Food (BLE) grant number 2814HS008. The funders had no role in study design, data collection and analysis, decision to publish, or preparation of the manuscript.

**Competing interests:** The authors have declared that no competing interests exist.

further light on the complex setting of associations a between parasitic diseases, milk yield and milk constituents, and management practices.

## Introduction

Parasitic diseases often represent a complex system with various consequences for the productivity and physiological integrity of the host. Globally, parasitic diseases rank among the most significant infectious diseases in ruminant livestock species [1, 2]. *Fasciola hepatica* and *Ostertagia ostertagi* represent the most abundant helminth species in dairy cattle around the world [3–5]. Infections in adult dairy cows have been associated with decreased animal health, impaired well-being, and compromised economic viability [6, 7]. Cows experience a reduction in milk yield, a decline in body condition, and poor reproductive performance [7–9]. For bovine fasciolosis, Schweizer et al. [10] have estimated financial losses of 299 € per infected cow in Switzerland. Furthermore, changes in milk composition such as lower milk fat and milk protein content have been linked to parasitic infections [11, 12].

Due to the complex nature of parasitic infections, including a large set of relevant factors as well as manifold associations with physiological integrity, health, and productivity of livestock, to determine which variable is outcome and which exposure is often not clearly possible. Cluster analysis is an unsupervised, heuristic, exploratory approach that identifies underlying patterns within the data and sorts the most similar observations into clusters that share common characteristics [13–15]. The basic idea is to aggregate data points within a cluster that are as similar as possible, whereas patterns between clusters are as different as possible. Unsupervised methods reduce subjective influence and show if and what kind of patterns are contained within the data. Such techniques may deliver insights which are not possible to obtain with a traditional, supervised modelling approach.

The objectives of the present study were (1) to explore if different clusters can be identified for farm-level bulk tank milk *F. hepatica* as well as *O. ostertagi* antibody status and milk parameters, and (2) to characterise potentially clustered farms and compare them in terms of external factors. We assumed that important associations exist among production parameters and antibody status that would naturally group farms in an unsupervised cluster analysis without a priori determination of target or predictor variables. These farms could subsequently be differentiated based on further criteria. We furthermore (3) intended to introduce a yet scarcely implemented modelling technique to the veterinary field which may represent a promising perspective for future investigations on complex biological systems. To our knowledge, this is the first study implementing an unsupervised machine learning technique in this context and the first time to use k–mode clustering and partitioning around medoids in veterinary epidemiology.

## Materials and methods

### Study farms

In an extensive, descriptive and cross-sectional study on dairy farms across Germany [16], data on housing conditions and animal health were collected. Dairy farms were located in three structurally and geographically different dairying regions in Germany. Within the three study regions North (federal states of Lower Saxony and Schleswig-Holstein), East (federal states of Thuringia, Saxony-Anhalt, Brandenburg, and Mecklenburg-Western Pomerania) and South (federal state of Bavaria), 765 farms (North: 253; East: 252; South: 260) with a total

number of 86,304 dairy cows (North: 24,980 cows; East: 49,936 cows South: 11,388 cows) were visited.

Farm selection process and sample calculation are elaborated on in [16, 17]. Briefly, different scenarios were calculated given a power of 80% and a level of significance of 5% in order to calculate an optimal and feasible sample size. Given these scenarios and considering feasibility, 250 farms were determined to be visited per study region. The selection of farms was assigned randomly and stratified on their administrative district and herd size within the federal state and study region. The national animal information data base (HIT) as well as farm data from the Milchprüfring Bayern e.V. provided information for sampling. Farms were randomly drawn from these data bases using an automated approach. A response rate of 30–40% was expected. Within each study region, a total amount of 1,250 farms, i.e. 5 times more farms than required for the study, were drawn from the underlying population in order to cover a response rate of at least 20%. Region–specific herd size cut-off values were determined in order to obtain a realistic distribution of herd sizes within the study population and due to structural differences in dairy farming in Germany [18].

Selected farms received an invitation letter to participate, containing information on the study procedure. The farm managers were asked to contact the regional study team on a voluntary basis. If they agreed to participate, they had to give their written consent for participation and data inspection. All farm-specific information was handled in accordance to the principles of the German and European data protection legislation. Study teams visited farms once between December 2016 and August 2019.

## On-farm data collection

Data collected during the farm visits were recorded via data entry forms and later manually transferred to a central SQL-data base. The individual ear tag number was recorded for each cow. All lactating and dry animals present on the day of the farm visit were subjected to body condition and locomotion scoring. Body condition score (BCS) was assessed following the 5-point system with 0.25 increments presented by Edmonson et al. [19]. A five-point locomotion scoring approach was implemented [20] to record abnormalities in posture and gait in loosely housed cows. In tie stall operations, cows underwent stall lameness scoring to document weight shifting between the rear limbs, sparing of a limb while standing, unequal weight bearing when stepping from side to side, and standing on the edge of the curb [21].

During the study period, a bulk tank milk (BTM) sample was collected from the central bulk tank on each farm by the farm manager to be analyzed for *F. hepatica* and *O. ostertagi* antibodies. To improve comparability across farms, farm managers were asked to collect the BTM sample towards the end of the grazing season, i.e. between August and October. Nevertheless, BTM samples taken in November ($n_{North}$ = 3, $n_{East}$ = 4, $n_{South}$ = 1) were also included in the analyses.

Data on milk yield (in kg), milk fat content (in %), milk protein content (in %), somatic cell count (SCC, in cells/ml), parity, and days in milk (DIM) were retrieved from HIT and the national milk recording system (DHI). In this context, data on milk yield and the contents of milk fat, milk protein, and SCC were available for up to 12 months prior to the farm visit. Information on pasture access and farming type (conventional vs. organic) was retrieved in a personal interview with the farm manager during the time of the farm visit and recorded via questionnaire. The farms were assigned to either 'tie stall operation', 'free stall operation' or 'other' if 80% of cows were housed in one of the two husbandry systems or another type of housing at the time of the visit.

## BTM *F. hepatica* and *O. ostertagi* antibody status

Exposure to *F. hepatica* and *O. ostertagi* on farm level was assessed by BTM antibodies determined with the IDEXX Fasciolosis Verification Test (IDEXX GmbH) and the Svanovir *O. ostertagi*–Ab ELISA (Boehringer Ingelheim) in a previous study [22]. As in the previous study and in accordance with the manufacturer's recommendation, *F. hepatica* ELISA results of S/P > 30% were considered seropositive, and for *O. ostertagi* ELISA results the threshold was set to ≥ 0.5 OD indicating herds likely to suffer from a negative impact on herd milk yield.

**Data editing and preparation.** The central SQL-database automatically checked plausibility of the data based on previously determined thresholds. Furthermore, two of the co–authors individually assessed the distribution and values of each variable, also in regard to other variables. If implausible values occurred, they were assessed both within the data base (to detect potential inconsistencies during data export) as well as within the original paper–based entry forms (to identify incorrect transfer of data from written records to database).

The statistical software R versions 4.0.3 and 4.2.0 [23] and R Studio [24] were used for all statistical analyses. A list of all R packages used in the current work is provided in S1 Table. For loosely housed cows, a cow was regarded as lame with a locomotion score ≥ 3 in accordance with previous work [25]. Tied cows were classified as lame if two out of the four behavioral patterns of the SLS were displayed during a 90 seconds period of observation [21, 26, 27].

As information on milk antibody status was retrieved on farm–level, animal–level data needed to be raised to farm level. This applied to the variables BCS, DIM, lameness, milk yield, milk fat content, milk protein content, somatic cell count, and parity. The farm level prevalence of lameness was calculated for each farm. It is important to note that information on milk yield, milk fat content, and milk protein content was available for each cow for a period of up to 12 months prior to the farm visit. Therefore, individual animals may have had a number of one up to twelve different values for each of these three parameters. To obtain a value that best reflected each of the three factors in the individual animal, a Bayesian non–parametric bootstrap approach with 1,000 resamples with replacements was implemented, which yielded Bayesian medians for all three variables. This allowed for a close reflection of the underlying values by the created estimates. Subsequently, a second round of bootstrapping with 1,000 resamples with replacements was conducted in order to transfer the animal–level information to farm-level median values which best reflected the on–farm situation in regard to the three factors.

Regarding the variables BCS, DIM, and parity, a single value was present for each individual animal. Hence, the information could be straightforwardly raised to farm level using the afore–mentioned Bayesian non–parametric bootstrap. This yielded a median value for BCS, DIM, and parity on farm level that most plausibly reflected the BCS and DIM situation as well as the parity level on each of the evaluated farms. A binary variable (*Fasciola*/*Ostertagia* seropositive/seronegative) was created based on the aforementioned thresholds of the BTM ELISAs.

## Cluster analysis

Farms with missing values were excluded from the analysis and a complete cases data set was generated. In addition to the binary variable 'antibody status' (*Fasciola* or *Ostertagia* seropositive/seronegative), each cluster analysis included the input variables milk yield, milk fat content, and milk protein content. Clustering was carried out separately for each of the study regions and each of the parasites.

Prior to cluster analysis, a distance measurement reflecting proximity or similarity across individual observations was required. As the input variables of the cluster analysis were of mixed nature (continuous variables milk yield, milk fat content, and milk protein content as

well as the categorical variable *Fasciola* or *Ostertagia* BTM status), Gower's distance, a common distance metric for a mix of categorical and continuous values and the first distance measure proposed in 1971 [28, 29], was computed as the average of partial dissimilarities across data points using the daisy() function from the cluster package [30]. The optimal number of clusters in the present study was determined via the silhouette method. The silhouette analysis evaluates the separation between resulting clusters, gives an indication of proximity between data points within a cluster and those in neighbouring clusters, and provides the optimal number of clusters [31].

The k-medoids algorithm applying partitioning around medoids (PAM) [32] was then implemented to infer similarities/dissimilarities from the Gower's distance matrix and to identify clusters of milk parameters and seropositivity/seronegativity for *F. hepatica* or *O. ostertagi*, respectively. Compared with the popular k-means algorithm which divides observations into a number of clusters, identifies *k* number of centroids (= means as the center of a cluster), and aggregates each observation to the cluster with the nearest mean [13, 14, 33], PAM replaces means with medoids, i.e. representatives of a cluster [34], since centroids as well as the Euclidean distance are not available for mixed data [35–37]. Data points are thus allocated based on certain similarities.

## Characterisation of clusters by means of random forest

External variables that had not been incorporated within the cluster analyses were used to compare clusters and to understand how these clusters and the farms within each of the clusters can be characterised and distinguished. The external variables included housing system, farm level lameness prevalence, farming type, herd size, pasture access, farm level somatic cell count, parity, DIM, sample month (month of BTM sample being submitted), sample year (year of BTM sample being submitted), visit month (month of farm visit), visit season (season of farm visit; spring: March–May; summer: June–August; autumn: September–November; winter: December–February), and visit year (year of farm visit). The biological reasoning behind these factors was expressed by a network structure (S1 Fig) created with the free software DAGitty (http://www.dagitty.net).

Variables were merged to the clustering data set. Farm level lameness prevalence was transformed into a categorical variable based on the variable's distribution within regions North (low prevalence < 14.72, medium prevalence 14.72–34.74, high prevalence > 34.74), East (low prevalence < 30.91, medium prevalence 30.91–47.92, high prevalence > 47.92), and South (low prevalence < 15.10%, medium prevalence 15.10–33.33%, high prevalence >33.33%). The number of cows being housed on each farm were sorted into three categories to reflect herd size within study region: North (small < 51.50 cows; medium 51.50–115.50 cows; large > 115.50 cows), East (small < 129.00; medium 129.00–418.00; large > 418.00), South (small < 27.00 cows; medium 27.00–59.00 cows; large > 59.00 cows).

The R package randomForest was used to implement Breiman's random forest algorithm for classification [38]. The mean decrease accuracy, describing how much removing a single variable reduces the accuracy of the model, was used as an indicator of variable importance. Hence, the variable with the highest importance gives the best prediction and contributes the most to the model fit and predictions compared with lower ranking variables [39]. This allowed to compare clusters 1 and 2 of each single cluster analysis by all external variables. ROC curves were generated for each random forest procedure. ROC curves are provided in S2 Fig. The r package rfPermute [40] was applied to perform a permutation test providing estimated p-values for the importance metric of the random forest.

## Results

### Descriptive results

Descriptive statistics for all study regions are summarised in Table 1.

Out of the 765 farms visited throughout the study period, 723 farms were enrolled to DHI and 646 farms submitted BTM samples. After merging the datasets for analysis and the removal of rows containing missing values, a total amount of 606 dairy farms were included in the current work.

**Region North.** Region North included a total number of 17,898 dairy cows on 191 farms. On 162 farms (84.82%), cows were housed in free stall facilities compared with tie stall barns and other housing systems (e.g. deep bedded packs) on 29 farms (15.18%). Organic farming principles were complied with on nine farms (4.7%), whereas 182 farms (95.29%) were conventionally managed. Pasture access was granted on 151 operations (79.06%) and absent on 40 farms (20.94%). Thirty farms (15.71%) were seropositive for *F. hepatica* and 91 farms for *O. ostertagi* (48.17%). Out of these farms, 26 farms were seropositive both for *F. hepatica* and *O. ostertagi* while four farms were seropositive only for *F. hepatica* and 65 farms only for *O. ostertagi*.

**Region East.** A total number of 201 farms with 24,980 cows was covered by the data set for region East. The vast majority of farms housed their cows in free stall facilities (n = 157, 78.11%). Cows had access to pasture on 107 operations (53.23%) and 20 farms (9.95%) were run on organic farming principles. Two BTM samples were seropositive for *F. hepatica* (1.00%) and 71 for *O. ostertagi* (35.32%).

**Region South.** The data set included 9,942 dairy cows on 214 farms. Out of the 214 farms, 55 housed their cows in tie stalls (25.70%) whereas 152 were free stall operations (71.03%) and seven were "other" (3.27%). A total amount of 181 farms (84.58%) were run conventionally compared with 33 organic farms (15.42%). Pasturing was carried out on 74 farms (34.58%).

**Table 1. Descriptive statistics of the data across the three study regions (North = 191 farms, East = 201 farms, South = 214 farms).**

| Variable | North | | | | | East | | | | | South | | | | |
|---|---|---|---|---|---|---|---|---|---|---|---|---|---|---|---|
| | Mean | Range | 1st Qu. | Median | 3rd Qu. | Mean | Range | 1st Qu. | Median | 3rd Qu. | Mean | Range | 1st Qu. | Median | 3rd Qu. |
| Herd size[1] | 93.71 | 10.00–486.00 | 51.50 | 79.00 | 115.50 | 334.80 | 1.00–2,821.00 | 129.00 | 245.00 | 418.00 | 46.46 | 5.00–231.00 | 27.00 | 40.50 | 59.00 |
| BCS[2] | 3.05 | 2.54–4.58 | 2.90 | 3.03 | 3.16 | 3.33 | 2.34–3.98 | 3.18 | 3.36 | 3.50 | 3.68 | 2.71–4.26 | 3.54 | 3.74 | 3.85 |
| Milk yield[2,3] | 26.08 | 20.00–30.82 | 24.83 | 26.08 | 27.48 | 25.65 | 14.74–31.78 | 24.61 | 26.10 | 27.14 | 25.19 | 19.40–31.20 | 24.21 | 25.45 | 26.50 |
| Milk fat[2,4] | 3.81 | 3.37–4.32 | 3.69 | 3.78 | 3.92 | 3.68 | 3.09–4.58 | 3.61 | 3.67 | 3.74 | 3.96 | 3.53–4.35 | 3.88 | 3.94 | 4.35 |
| Milk protein[2,4] | 3.20 | 2.93–3.49 | 3.12 | 3.18 | 3.29 | 3.11 | 2.54–3.49 | 3.07 | 3.13 | 3.16 | 3.35 | 3.12–3.61 | 3.30 | 3.36 | 3.40 |
| SCC[2,5,6] | 217.40 | 122.90–663.90 | 183.90 | 221.70 | 239.20 | 228.27 | 27.64–365.94 | 199.49 | 222.58 | 254.36 | 205.20 | 106.2–421.8 | 167.0 | 197.7 | 233.0 |
| Lameness[7] | 25.82 | 0.00–76.92 | 14.72 | 23.08 | 34.74 | 38.51 | 0.00–77.63 | 30.91 | 39.00 | 47.92 | 25.59 | 0.00–76.47 | 15.10 | 23.57 | 33.33 |
| DIM[2] | 213.40 | 131.60–360.70 | 194.60 | 209.40 | 231.40 | 205.20 | 101.30–333.70 | 187.80 | 204.50 | 219.20 | 197.20 | 118.50–614.70 | 175.50 | 191.00 | 197.20 |
| Parity[2] | 2.85 | 1.79–5.10 | 2.60 | 2.77 | 3.08 | 2.80 | 1.86–5.15 | 2.54 | 2.72 | 2.97 | 2.90 | 1.85–4.76 | 2.57 | 2.85 | 3.11 |

[1] number of lactating and dry cows

[2] bayesian median value per farm

[3] in kg/day

[4] in %

[5] × 1,000

[6] in number of cells/ml

[7] Farm level prevalence in %

BTM samples from 51 farms were positive for *F. hepatica* (24.83%) and 79 farms for *O. ostertagi* antibodies (36.92%). Six farms were positive only for *F. hepatica*, and 34 farms only for *O. ostertagi*. Forty–five farms were positive for both parasites.

## Cluster analyses

Clustering was performed separately for each of the region and each parasite. Cluster analyses for *O. ostertagia* were conducted in all study regions, while *F. hepatica* was evaluated in study regions North and South. In region East, only two farms were positive for *F. hepatica*, therefore cluster analyses were not conducted for this parasite in region East. It is important to understand that based on the aforementioned, two clusters (a cluster 1 and a cluster 2) were present in each of the analyses. Cluster analyses were entirely mutually independent, e.g. cluster 1 in the *F. hepatica* analysis in region North represents different outcomes compared with cluster 1 in region South.

For all cluster analyses, the silhouette method selected two clusters to be the optimal number of clusters to group the data points, i.e. the farms, in alignment with the underlying data (S3 Fig).

**F. hepatica cluster analyses.**   Fig 1 contains two cluster plots visualising the allocation of farms to the presented clusters for the *F. hepatica* analysis in study regions North and South.

As for region North, 161 farms (84.29%) were assigned to cluster 1 and 30 farms (15.71%). All 30 farms in cluster 2 were positive for *F. hepatica* and grouped together with their respective production parameters. Descriptive cluster statistics for the *F. hepatica* cluster analyses in study regions North and South are displayed in Table 2 for continuous variables and in Table 3 for categorical variables, respectively.

In region South, cluster 1 included 51 farms (23.83%) compared with 163 farms (76.17%) in cluster 2. Similarly to region North, all 51 farms positive for *F. hepatica* grouped together with their respective production parameters in one cluster (cluster 1).

For both regions, the presence of pasture access appears to discriminate between clusters. In region North cluster 1 incorporated fewer farms (75.16%) providing pasture access to their animals compared with farms in cluster 2 (100%). In study region South, pasture access appeared to be more common in farms within cluster 1 (92.16%) compared with cluster 2 (83.44%). Another predictor differing by clusters in both regions was farming type: conventional farming was more prevalent among Northern farms in cluster 1 (96.89%) than in cluster 2 (86.67%). Among farms in region South, more organic farms were allocated in cluster 1 (41.18%) compared with cluster 2 (7.36%). Moreover, positivity for *O. ostertagi* appeared to be different between clusters in both regions. In region North, the majority (86.67%) of *O ostertagi* positive farms were assigned to cluster 2 compared with 40.37% in cluster 1. Similarly, in region South, 88.24% of farms positive for *O. ostertagi* were within one cluster (cluster 1), whereas only 20.86% of positive farms were part of the other cluster (cluster 2).

**O. ostertagi cluster analyses.**   The results from the clustering procedure for *O. ostertagi* (all study regions) are illustrated in Fig 2.

In study region North, all 91 farms positive for *O. ostertagi* were grouped into cluster 1 whereas the remaining 100 farms were grouped into cluster 2. Similarly in the other two study regions, positive farms clustered together: in region East all 71 *O. ostertagi*–positive farms were included in cluster 1 and the remaining 130 farms in cluster 2. In region South, all 79 farms positive for *O. ostertagi* were assigned to cluster 1 and the 135 negative farms to cluster 2.

Table 4 provides a summary of the cluster descriptive statistics across study regions for continuous variables, whereas results for categorical variables are provided in Table 5.

The presence of pasture access appeared to differ among clusters in all study regions: pasture access was more prevalent in one cluster than in the other cluster across study regions.

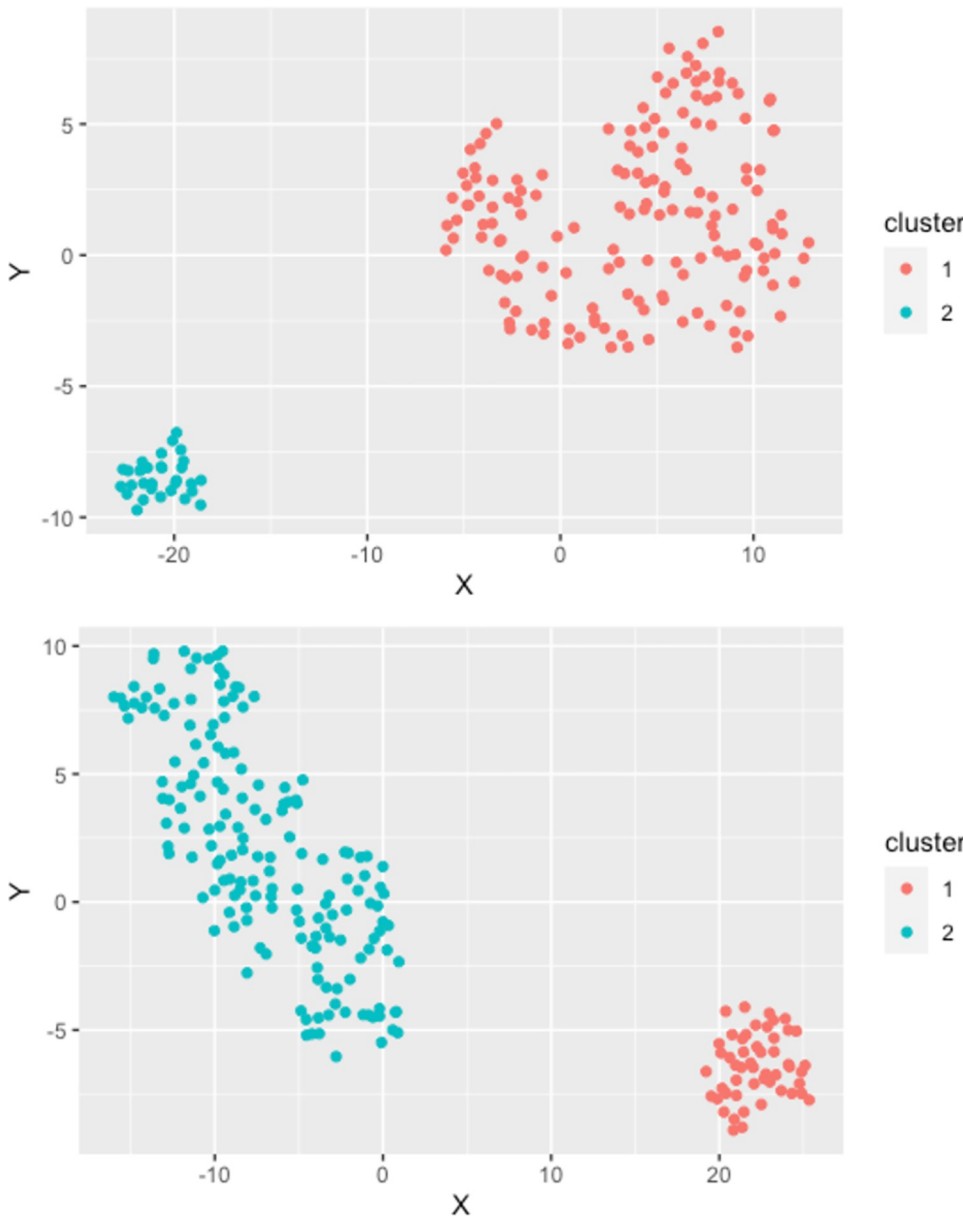

**Fig 1. Cluster plot of the partitioning around medoids clustering process for *F. hepatica* (regions North and South).** Region North (top): Two distinct clusters are displayed with 161 farms in cluster 1 (red) and 30 observations in cluster 2 (blue). Region South (bottom): Two clusters with 51 observations in cluster 1 (red) and 163 observations in cluster 2 (blue) naturally aggregated.

Similarly, more organic farms were assigned to one cluster in all study regions compared with the other cluster. Positivity for. *F. hepatica* did not appear to as clearly distinguish between clusters as positivity for *O. ostertagi* did in *the F. hepatica* cluster analyses.

### Classification of clusters by means of random forest

*F. hepatica* **analyses.** Fig 3 illustrates the outcome of the random forest classification of the *F. hepatica* cluster analyses for study regions North (Fig 3A) and South (Fig 3B).

**Table 2. Descriptive cluster statistics of the *F. hepatica* cluster analysis in regions North and South (continuous variables).**

| | Region North | | | | | Region South | | | | |
|---|---|---|---|---|---|---|---|---|---|---|
| | Cluster 1 | | | | | Cluster 1 | | | | |
| Variable | Mean | Range | 1st Qu. | Median | 3rd Qu. | Mean | Range | 1st Qu. | Median | 3rd Qu. |
| BCS | 3.06 | 2.54–4.58 | 2.93 | 3.05 | 3.16 | 3.55 | 2.71–4.07 | 3.34 | 3.64 | 3.78 |
| SCC[1,2] | 218.90 | 122.90–663.90 | 184.30 | 211.80 | 242.70 | 210.40 | 112.50–393.40 | 175.20 | 205.50 | 224.50 |
| Lame[3] | 25.05 | 0.00–76.92 | 15.71 | 23.08 | 34.75 | 15.98 | 0.00–48.00 | 7.87 | 14.81 | 21.86 |
| DIM | 213.50 | 143.50–360.70 | 197.30 | 210.30 | 231.10 | 200.10 | 139.70–291.30 | 174.60 | 201.80 | 219.70 |
| Parity | 2.85 | 1.79–5.06 | 2.60 | 2.77 | 3.09 | 3.11 | 2.13–4.76 | 2.74 | 2.96 | 3.47 |
| Milk yield[4] | 26.24 | 20.77–30.31 | 24.95 | 26.36 | 27.60 | 24.59 | 21.12–29.23 | 23.00 | 24.48 | 25.60 |
| Milk fat[5] | 3.79 | 3.37–4.24 | 3.68 | 3.76 | 3.91 | 3.92 | 3.53–4.18 | 3.83 | 3.92 | 4.00 |
| Milk protein[5] | 3.19 | 2.93–3.49 | 3.11 | 3.17 | 3.27 | 3.33 | 3.18–3.61 | 3.27 | 3.32 | 3.39 |
| | Cluster 2 | | | | | Cluster 2 | | | | |
| BCS | 2.98 | 2.73–3.41 | 2.82 | 2.94 | 3.08 | 3.73 | 2.92–4.26 | 3.62 | 3.77 | 3.87 |
| SCC[1,2] | 209.80 | 134.40–342.00 | 182.30 | 211.00 | 228.10 | 203.60 | 106.20–421.80 | 164.60 | 194.50 | 235.20 |
| Lame[3] | 24.56 | 0.00–73.13 | 13.40 | 22.57 | 33.77 | 28.60 | 0.00–76.47 | 19.59 | 26.32 | 36.24 |
| DIM | 212.70 | 131.60–360.60 | 183.80 | 202.70 | 233.70 | 196.30 | 118.50–614.70 | 176.70 | 189.90 | 210.80 |
| Parity | 2.85 | 1.90–4.15 | 2.61 | 2.77 | 3.07 | 2.84 | 1.85–4.50 | 2.54 | 2.81 | 3.07 |
| Milk yield[4] | 25.27 | 20.00–30.82 | 24.43 | 25.26 | 26.07 | 25.38 | 19.40–31.20 | 24.41 | 25.66 | 25.55 |
| Milk fat[5] | 3.92 | 3.62–4.32 | 3.81 | 3.90 | 4.04 | 3.97 | 3.69–4.35 | 3.88 | 3.96 | 4.05 |
| Milk protein[5] | 3.26 | 3.07–3.46 | 3.18 | 3.25 | 3.36 | 3.36 | 3.18–3.53 | 3.31 | 3.36 | 3.40 |

[1] × 1,000

[2] in number of cells/ml

[3] Farm level prevalence in %

[4] in kg

[5] in %

BTM positivity for *O. ostertagi* (North: p = 0.01; South: p = 0.01) and pasture access (North: p = 0.050; South: p = 0.01) were the two most important variables discriminating clusters both in study region North and South. Whereas farming type discriminated among clusters ranking third in region North (p = 0.042), the variable ranked fourth in region South (p = 0.01). The third most important variable in region South was farm level lameness prevalence (p = 0.01). Furthermore, herd size appeared to be relevant in discriminating among clusters in region South (p = 0.01).

***O. ostertagi* analyses.** As for the *O. ostertagi* analyses, eight variables appeared to be the most important discriminating clusters in region North (Fig 4A): lameness prevalence (p = 0.01), BTM status for *F. hepatica* (p = 0.01), BCS (p = 0.01), herd size (p = 0.01), pasture access (p = 0.01), SCC (p = 0.01), DIM (p = 0.04), and farming type (p = 0.02). In region East, three variables, i. e. farming type (p = 0.01), pasture (p = 0.01), and DIM (p = 0.04) were the most important criteria differentiating both clusters (Fig 4B). BTM status for *F. hepatica* (p = 0.01), pasture access (p = 0.01), and farming type (p = 0.01) were the top ranking variables in region South (Fig 4C). Since only two farms were positive for *F. hepatica* in region East, the variable was excluded from the random forest approach in this study region.

## Discussion

The aims of the present study were (1) to evaluate if distinct groups of farms can be identified in regard to milk parameters and BTM antibodies against two major helminth endoparasites in dairy cattle by means of an exploratory, unsupervised machine learning approach and (2) to

**Table 3. Descriptive statistics (observations per cluster) of the _F. hepatica_ cluster analysis in study regions North and South (categorical variables).**

| Study Region Cluster (Counts [%]) | Variable Categories Counts [%] | | | | | | | | | | | | | |
|---|---|---|---|---|---|---|---|---|---|---|---|---|---|---|
| | Housing system | | | Pasture access | | Farming type | | Herd size[1] | | | _F. hepatica_ | | _O. ostertagi_ | |
| | Tie Stall | Free Stall | Other | Yes | No | Conventional | Organic | Small | Medium | Large | Negative | Positive | Negative | Positive |
| Region North | | | | | | | | | | | | | | |
| Cluster 1 (161.00 [84.29]) | 7.00 [4.35] | 137.00 [85.09] | 17.00 [10.56] | 121.00 [75.16] | 40.00 [24.84] | 156.00 [96.89] | 5.00 [3.11] | 41.00 [25.47] | 79.00 [49.07] | 41.00 [25.47] | 161.00 [100] | 0.00 [0.00] | 96.00 [59.63] | 65.00 [40.37] |
| Cluster 2 (30.00 [15.71]) | 1.00 [3.33] | 25.00 [83.33] | 4.00 [13.33] | 30.00 [100.00] | 0.00 [0.00] | 26.00 [86.67] | 4.00 [13.33] | 7.00 [23.33] | 16.00 [53.33] | 7.00 [23.33] | 0.00 [0.00] | 30.00 [100.00] | 4.00 [13.33] | 26.00 [86.67] |
| Region South | | | | | | | | | | | | | | |
| Cluster 1 (51.00 [23.83]) | 22.00 [43.14] | 28.00 [54.90] | 1.00 [1.96] | 47.00 [92.16] | 4.00 [7.84] | 30.00 [58.82] | 21.00 [41.18] | 23.00 [45.10] | 26.00 [50.98] | 2.00 [3.93] | 0.00 [0.00] | 51.00 [100.00] | 6.00 [11.76] | 45.00 [88.24] |
| Cluster 2 (163.00 [76.17]) | 33.00 [20.25] | 124.00 [76.07] | 6.00 [3.68] | 136.00 [83.44] | 27.00 [16.56] | 151.00 [92.64] | 12.00 [7.36] | 29.00 [17.79] | 84.00 [51.53] | 50.00 [30.67] | 163.00 [100.00] | 0.00 [0.00] | 129.00 [79.14] | 34.00 [20.86] |

[1] number of cows present on farm

region North: small < 51.50 cows, medium 51.50–115.50 cows, large > 115.50 cows

region South: small < 27 cows, medium 27–59 cows, large > 59 cow

compare potential clusters based on external factors. Furthermore, we intended to introduce k–medoids clustering and partitioning around medoids to a veterinary context.

As we assumed that a certain number of groups may be present within the data, cluster analysis was chosen to reveal these groups in an unsupervised way. This assumption was based on previous work on the association of production parameters with the presence of _F. hepatics_ and _O. ostertagi_ [7–9, 42]. Cluster analysis is an exploratory technique that autonomously identifies potential naturally present similarities among observations within an underlying data set and subsequently assigns data points to different clusters [13–15]. The k-medoids algorithm was chosen for clustering as it allows for the evaluation of mixed data. Furthermore, it has the advantage of robustness and is not affected by potential outliers or extreme values [32, 34, 41]. To our knowledge, this is the first study implementing a cluster analysis approach to evaluate production parameters and anti-parasite BTM antibody status in dairy cows. More-over, the current work is among the very first of its kind to implement a k-medoids cluster algorithm and PAM in epidemiology [42, 43]. Clustering revealed two distinct clusters for all analyses as assumed. We hence can infer that similarities exist between farms regarding the input variables (milk yield, milk fat content, milk protein content, antibody status) of the pres-ent analysis. The formation of distinct clusters of _F. hepatica_ and _O. ostertagi_ BTM results and parameters of milk production (i.e. milk yield) and milk quality (i.e. milk fat content, milk pro-tein content) in the current study supports previous findings on associations of parasite expo-sure with production parameters [9, 44, 45]. Interestingly, all parasite–positive farms grouped together with their respective production parameters across all analyses. Hence, seropositivity for _F. hepatica_ or _O. ostertagi_ was a principal feature characterising clusters, as apparently sero-positive farms strongly grouped together with the respective production parameters and

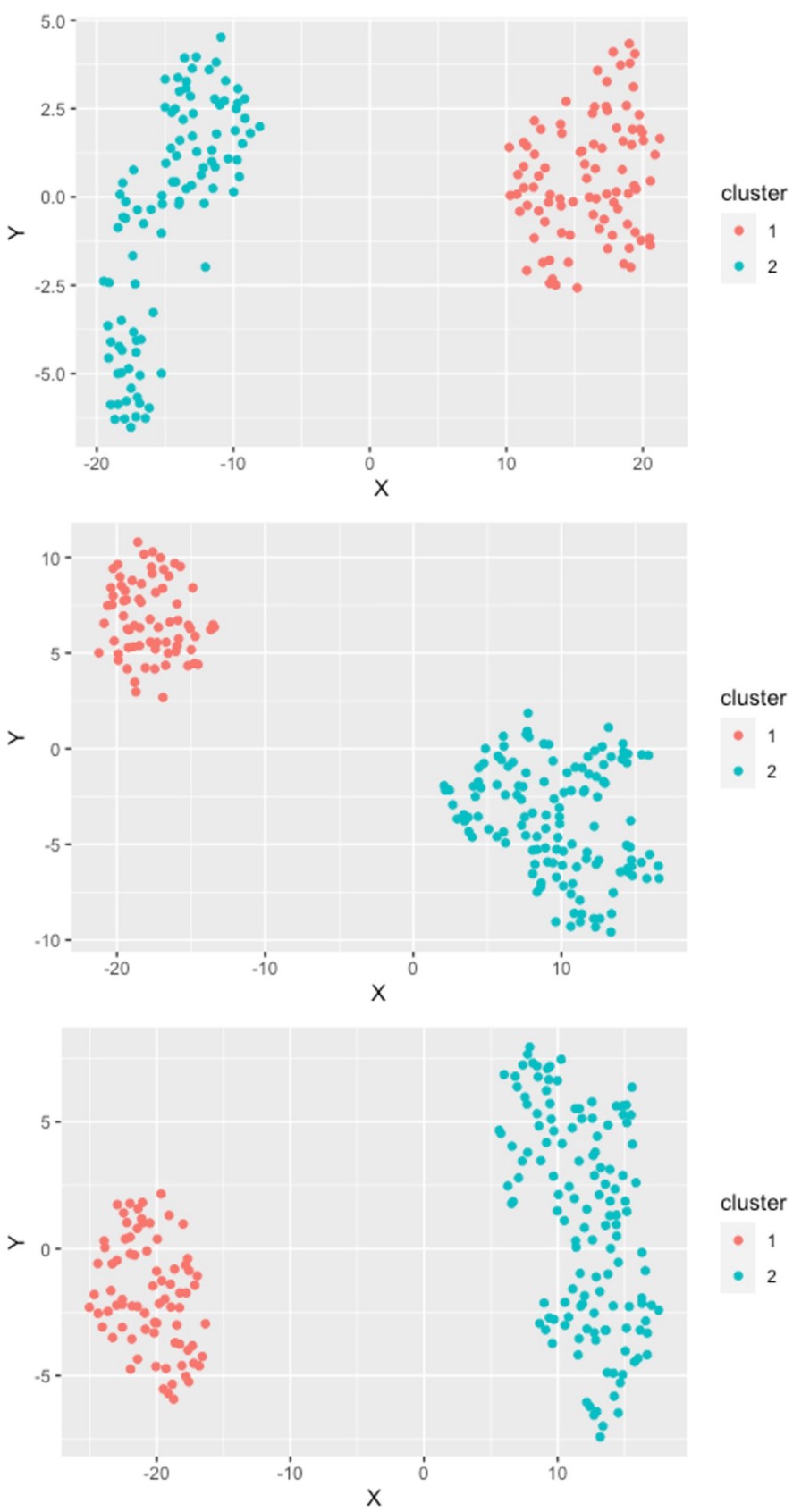

**Fig 2. Cluster plot of the k-medoids clustering process for *O. ostertagi* in the three study regions.** Region North (top): Two clusters with 91 observations in cluster 1 (red) and 100 observations in cluster 2 (blue). Region East (middle): Two clusters with 71 observations in cluster 1 (red) and 130 observations in cluster 2 (blue). Region South (bottom): Two clusters with 79 observations in cluster 1 (red) and 135 observations in cluster 2 (blue).

formed a separate cluster compared with seronegative farms. Furthermore, this finding not only corroborates the results of the present study that relevant differences exist among seropositive and seronegative farms per se, but also indicates that marked differences exist between those farms in regard to production parameters. Associations between parasite exposure and milk composition have been previously described [7–9]. The results of distinct groups of production parameters and BTM antibody status for *F. hepatica* and *O. ostertagi* in the current study appear intuitive and plausible and confirm the evidence from previous work [46–48]. May et al. [49] estimated an average loss of 1.62 kg milk per cow and day in dairy herds strongly infected with *F. hepatica* based on BTM ELISA, similar to the current work. The authors furthermore showed negative effects of *F. hepatica* infection status on milk quality as represented by milk protein and milk fat. For *O. ostertagi*, the situation appears similar to the

**Table 4. Descriptive statistics of the *O. ostertagi* cluster analysis across study regions (continuous variables).**

| | Region North | | | | | Region East | | | | | Region South | | | | |
|---|---|---|---|---|---|---|---|---|---|---|---|---|---|---|---|
| | Cluster 1 | | | | | Cluster 1 | | | | | Cluster 1 | | | | |
| Variable | Mean | Range | 1st Qu. | Median | 3rd Qu. | Mean | Range | 1st Qu. | Median | 3rd Qu. | Mean | Range | 1st Qu. | Median | 3rd Qu. |
| BCS | 3.02 | 2.54–4.58 | 2.85 | 2.97 | 3.12 | 3.28 | 2.34–3.93 | 3.11 | 3.33 | 3.49 | 3.60 | 2.71–4.07 | 3.44 | 3.67 | 3.80 |
| SCC[1,2] | 224.10 | 134.40–663.90 | 183.70 | 211.80 | 244.90 | 232.44 | 27.64–365.94 | 198.32 | 230.94 | 267.71 | 206.60 | 106.20–393.40 | 170.80 | 205.50 | 227.40 |
| Lame[3] | 22.14 | 0.00–76.92 | 10.19 | 19.12 | 28.00 | 36.52 | 0.00–70.95 | 25.40 | 38.12 | 49.07 | 19.92 | 0.00–57.90 | 9.55 | 18.75 | 26.77 |
| DIM | 211.40 | 131.60–360.60 | 187.80 | 205.00 | 230.20 | 210.40 | 101.30–333.70 | 193.90 | 208.40 | 226.40 | 201.40 | 118.50–291.30 | 176.30 | 203.00 | 220.90 |
| Parity | 2.90 | 1.79–5.10 | 2.59 | 2.78 | 3.18 | 2.82 | 1.97–5.09 | 2.52 | 2.71 | 3.01 | 3.00 | 1.92–4.76 | 2.59 | 2.92 | 3.21 |
| Milk yield[4] | 25.52 | 20.00–29.90 | 24.42 | 25.50 | 26.71 | 24.40 | 14.74–29.49 | 23.25 | 24.92 | 26.11 | 24.58 | 20.11–31.20 | 22.89 | 24.40 | 26.03 |
| Milk fat[5] | 3.84 | 3.37–4.32 | 3.72 | 3.79 | 3.96 | 3.71 | 3.40–4.41 | 3.63 | 3.70 | 3.77 | 3.92 | 3.53–4.18 | 3.84 | 3.92 | 4.00 |
| Milk protein[5] | 3.21 | 2.95–3.49 | 3.13 | 3.19 | 3.30 | 3.10 | 2.76–3.49 | 3.04 | 3.11 | 3.16 | 3.34 | 3.18–3.61 | 3.27 | 3.33 | 3.40 |
| | Cluster 2 | | | | | Cluster 2 | | | | | Cluster 2 | | | | |
| BCS | 3.07 | 2.69–3.81 | 2.97 | 3.06 | 3.17 | 3.35 | 2.84–3.98 | 3.21 | 3.37 | 3.50 | 3.74 | 2.92–4.26 | 3.62 | 3.79 | 3.87 |
| SCC[1,2] | 211.40 | 122.90–458.60 | 184.30 | 211.00 | 234.80 | 226.00 | 132.00–332.40 | 200.80 | 220.30 | 246.90 | 204.50 | 115.80–421.80 | 166.60 | 194.50 | 236.40 |
| Lame[3] | 29.17 | 3.18–61.36 | 17.67 | 27.48 | 39.10 | 39.60 | 0.00–77.63 | 32.64 | 39.40 | 47.59 | 28.90 | 0.00–76.47 | 19.59 | 27.12 | 36.47 |
| DIM | 215.20 | 166.60–360.70 | 199.10 | 212.20 | 232.60 | 202.30 | 127.00–320.10 | 185.00 | 203.80 | 217.20 | 194.70 | 134.50–614.70 | 175.20 | 188.90 | 203.90 |
| Parity | 2.80 | 1.98–4.18 | 2.60 | 2.74 | 2.99 | 2.79 | 1.86–5.15 | 2.56 | 2.74 | 2.95 | 2.84 | 1.85–4.50 | 2.56 | 2.82 | 3.07 |
| Milk yield[4] | 26.60 | 20.77–30.82 | 25.43 | 26.75 | 27.84 | 26.33 | 16.36–31.78 | 25.48 | 26.51 | 27.58 | 25.55 | 19.40–29.11 | 24.76 | 25.75 | 26.59 |
| Milk fat[5] | 3.78 | 3.44–4.13 | 3.68 | 3.76 | 3.91 | 3.66 | 3.09–4.58 | 3.60 | 3.66 | 3.72 | 3.98 | 3.69–4.35 | 3.89 | 3.97 | 4.05 |
| Milk protein[5] | 3.29 | 2.93–3.45 | 3.11 | 3.18 | 3,28 | 3.12 | 2.54–3.33 | 3.08 | 3.13 | 3.16 | 3.36 | 3.19–3.53 | 3.31 | 3.36 | 3.40 |

[1] × 1,000

[2] in number of cells/ml

[3] Farm level prevalence in %

[4] in kg

[5] in %

**Table 5. Descriptive statistics (observations per cluster) of the *O. ostertagi* cluster analysis across study regions (categorical variables).**

| Study Region Cluster (Counts [%]) | Variable Categories Counts [%] | | | | | | | | | | | | | |
| --- | --- | --- | --- | --- | --- | --- | --- | --- | --- | --- | --- | --- | --- | --- |
| | Housing system | | | Pasture access | | Farming type | | Herd size[1] | | | *F. hepatica* | | *O. ostertagi* | |
| | Tie Stall | Free Stall | Other | Yes | No | Conventional | Organic | Small | Medium | Large | Negative | Positive | Negative | Positive |
| Region North | | | | | | | | | | | | | | |
| Cluster 1 (91.00 [47.64]) | 7.00 [7.69] | 70.00 [76.92] | 14.00 [15.38] | 85.00 [93.41] | 6.00 [6.59] | 82.00 [90.11] | 9.00 [9.89] | 27.00 [29.67] | 54.00 [59.34] | 10.00 [10.99] | 65.00 [71.43] | 26.00 [28.57] | 0.00 [0.00] | 91.00 [100.00] |
| Cluster 2 (100.00 [52.36]) | 1.00 [1.00] | 92.00 [92.00] | 7.00 [7.00] | 66.00 [66.00] | 34.00 [34.00] | 100 [100.00] | 0.00 [0.00] | 21.00 [21.00] | 41.00 [41.00] | 38.00 [38.00] | 96.00 [96.00] | 4.00 [4.00] | 100.00 [100.00] | 0.00 [0.00] |
| Region East | | | | | | | | | | | | | | |
| Cluster 1 (71.00 [35.32]) | 2.00 [2.82] | 55.00 [77.46] | 14.00 [19.72] | 55.00 [77.46] | 16.00 [22.54] | 55.00 [77.46] | 16.00 [22.54] | 24.00 [33.80] | 38.00 [53.52] | 9.00 [12.68] | - | - | 0.00 [0.00] | 71.00 [100.00] |
| Cluster 2 (130.00 [64.68]) | 0.00 [0.00] | 102.00 [78.46] | 28.00 [21.54] | 78.00 [60.00] | 52.00 [40.00] | 126.00 [96.92] | 4.00 [3.08] | 26.00 [20.00] | 63.00 [48.46] | 41.00 [31.54] | - | - | 130.00 [100.00] | 0.00 [0.00] |
| Region South | | | | | | | | | | | | | | |
| Cluster 1 (79.00 [36.92]) | 27.00 [34.18] | 50.00 [63.29] | 2.00 [2.53] | 57.00 [72.15] | 22.00 [27.85] | 52.00 [65.82] | 27.00 [34.18] | 27.00 [34.18] | 39.00 [49.37] | 13 [16.46] | 34.00 [43.04] | 45.00 [56.96] | 0.00 [0.00] | 79.00 [100.00] |
| Cluster 2 (135.00 [63.08]) | 28.00 [20.74] | 102.00 [75.56] | 5.00 [3.70] | 17.00 [12.59] | 118.00 [87.41] | 129.00 [95.56] | 6.00 [4.44] | 25 [18.52] | 71 [52.59] | 39 [28.89] | 129.00 [95.56] | 6.00 [4.44] | 135.00 [100.00] | 0.00 [0.00] |

[1] number of cows present on farm, categorised

region North: small < 51.50 cows, medium 51.50–115.50 cows, large > 115.50 cows

region East: small < 129.00 cows, medium 129.00–418.00 cows, large > 418.00 cows

region South: small < 27 cows, medium 27–59 cows, large > 59 cow

evidence available for *F. hepatica*. Blanco-Penedo et al. [50] have reported an adverse effect of BTM infection status on milk yield, which is underscored by further studies indicating a strong association between *O. ostertagi* antibody status and milk production and quality parameters [51–53]. Interestingly, in their recent study which in large parts was based on the data set collected in the underlying three years cross–sectional study throughout Germany, Springer et al. [22] identified a negative impact on milk production level only in high–performance dairy breeds while lower milk fat content was observed in dual–purpose herds positive for *O. ostertagi*. The authors traced it back to region–specific characteristics which may be relevant in this context. They also speculated that dual–purpose breeds may be more resilient towards the adverse effects of parasitic infection compared with high performance dairy breeds. In the current study, breed was not included in the data set as the main focus was different. Furthermore, Springer et al. [22] built generalised linear models and results may differ as a result of the chosen technique. The implementation of an unsupervised machine learning approach in the present study is a particular advantage in comparison with common supervised modelling techniques, since no prior specification is made about response and explanatory variables and

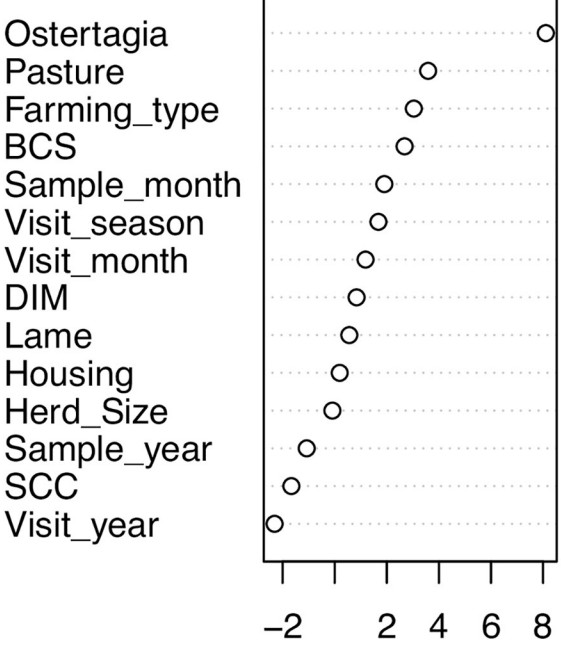

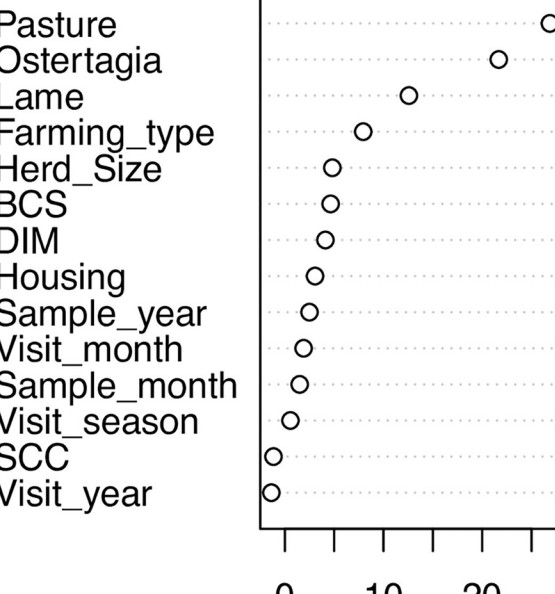

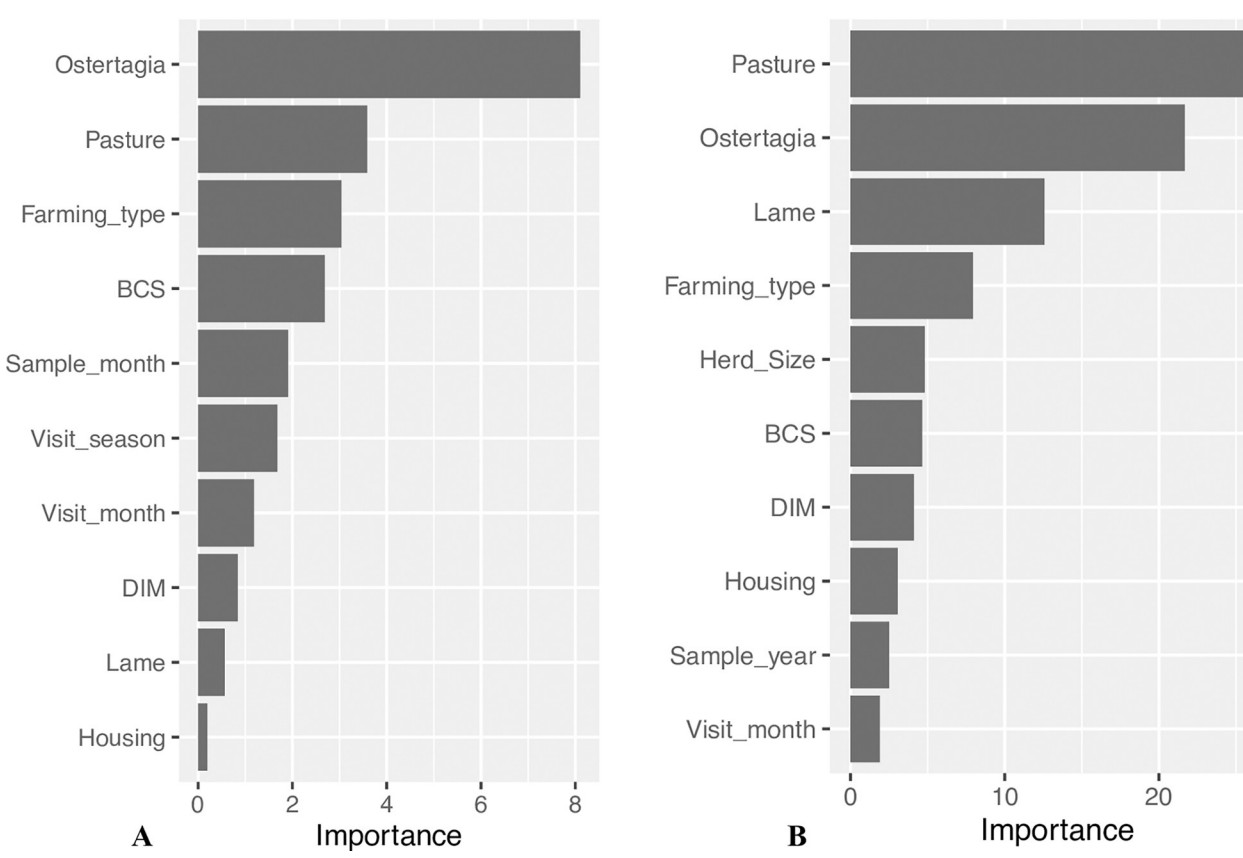

**Fig 3. Mean decrease accuracy plots and variable importance plots for the random forest classification process of clusters 1 and 2 in study regions North and South (*F. hepatica* analyses).** A: study region North. B: study region South. The mean decrease accuracy plot expresses how much accuracy the model loses by excluding each variable. The more the accuracy suffers, the more important the variable is in distinguishing clusters. Vice versa, the higher the value of the mean decrease accuracy, the higher the importance of the variable in the model. The first three criteria appear to be the most valuable ones in characterizing clusters in region North, whereas five variables were ranked as the most relevant in region South.

each factor equally enter the analysis. It is yet important to be aware that cluster analysis is exploratory and does not explain the quality of similarity within as well as the quality of dissimilarity between the clusters. Therefore, further steps are required following a cluster analysis in order to compare formed clusters and to identify external (not part of the cluster analysis) variables, which characterise clusters. The random forest algorithm was implemented in the present study to identify factors distinguishing clusters for the single analyses. As two clusters were present in each of the cluster analyses, cluster was used as a binary outcome for the random forest approach. Random forest consistently provide among the highest accuracy of prediction compared with other classification techniques and complex random forest classifiers produce high discriminative performance [54, 55]. Throughout the cluster analyses, the

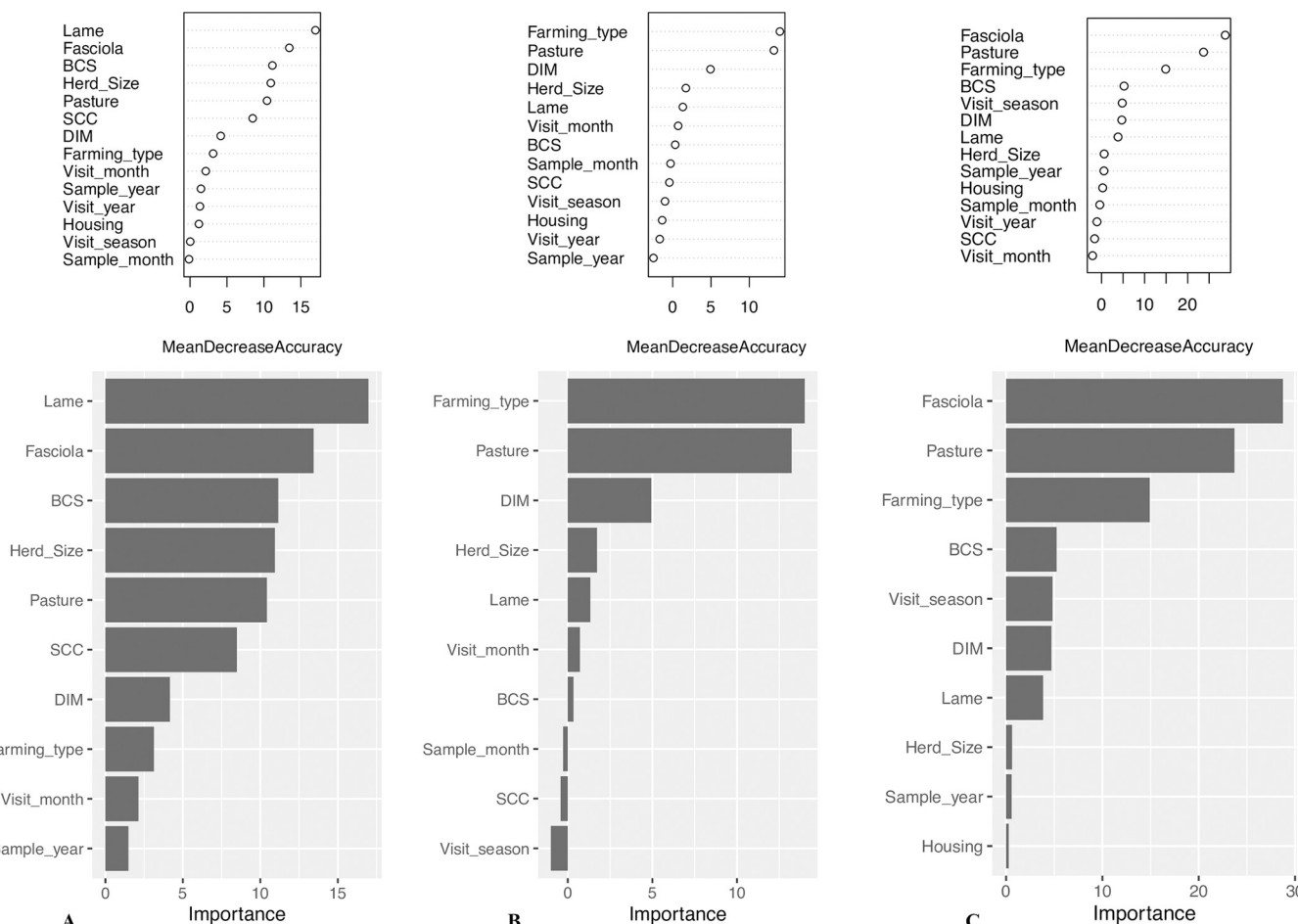

**Fig 4. Mean decrease accuracy plots and variable importance plots for the random forest classification process of clusters 1 and 2 in study regions North and South (*O. ostertagi* analyses).** A: study region North. B: study region South. C: study region South. Changes of prediction accuracy, i. e. decrease of prediction accuracy if excluding single variables. In region North the first eight factors are the most valuable ones characterising clusters compared with the remaining variables. In study regions East and South, the first three variables were the most important ones differentiating clusters.

random forest algorithm confirmed that organic farming and pasture access were relevant factors differentiating clusters of similar quality both for *F. hepatica* and *O. ostertagi*. In all random forests, these two factors were among the top ranking criteria of classification. Solely in the *O. ostertagi* cluster analysis of region North, pasture access (ranking fifth) and organic farming (ranking eight), despite being identified relevant based on the permutation test, were not ranking at the very top of variables discriminating clusters. This is probably due to the fact that compared with the remaining two study regions, pasture access is more prevalent on farms in region North and hence other criteria appeared to be more relevant to differ between farms. Organic farming is based on the idea of maintaining animal health and productivity through proactive rather than reactive management [56]. Practices often differ between organic and conventional operations [57, 58]. Dairy cow health and welfare represent one of the main components of an organic farming mindset and are achieved by improved housing conditions, less intensive production, and smaller herds. Furthermore, as organic farming associations often impose strict rules on the provenance and usage of feed components, the proportion of organic concentrate is limited and feeding regimes frequently are less intensive in an organic setting. Most importantly, pasturing cows is a central element of organic dairy farming [56–58]. Based on the epidemiology of *F. hepatica* and *O. ostertagi*, the results of the random forests regarding organic farming and pasture access are plausible findings. Feeding grass is a crucial element in the transmission dynamics of endoparasitic helminths in cattle. Infective stages present on pasture are transmitted to potential hosts via fresh grass, hay or silage [59–61]. Moreover, the aforementioned differences between farming types may also explain the difference of the clusters in regard to the other input variable (milk yield, milk fat, milk protein). The milk composition related input variables of the cluster analyses may well be influenced by the varying settings (e.g. pasture access, feeding regimes) on conventional and organic farms.

BTM positivity for *F. hepatica* was an important factor differentiating clusters in regions North and South of the *O. ostertagi* analysis. Similarly, BTM positivity for *O. ostertagi* appeared to be top ranking when discriminating clusters in the *F. hepatica* analyses. This is supported by the descriptive results that co–infections with both parasites was very common throughout study regions. Given the biology and life cycle of the parasites which involve transmission through oral ingestion of infective stages via grass or pasture, it appears plausible that they might both benefit from similar factors present at farm level and co–infections may be common [62]. However, it is very interesting to understand that not only is co–infection very frequent, but also that the presence of the one parasite appeared to characterise the presence of the other parasite. It would be a promising approach for future investigation to evaluate, how both species are intertwined in their impact on dairy farms. Furthermore, identifying criteria that farms with co–infection share among each other as well as criteria that differ between co-infected, mono–infected, and parasite–free operations may yield practically relevant insights. Knowledge on infection intensity, impact, spatial distribution, and risk factors are pivotal to develop targeted intervention strategies.

Interestingly, lameness prevalence appeared to be a relevant variable discriminating clusters in the *F. hepatica* analysis of region South as well as in the *O. ostertagi* analysis of Region North. In the latter, lameness prevalence was the most important criterion to classify clusters. Lameness is a very common, major problem in dairy production that impinges upon animal welfare, physiological integrity, and economic viability [17, 63, 64]. Lame animals are not only impaired in their natural behavioral patterns but also experience severe, often chronic pain [64, 65]. The association of lameness with production related parameters has been demonstrated by previous research [17, 25, 66, 67]. It is hence reasonable to assume that lameness also has an association with the milk composition parameters used as input variables in the

current cluster analysis. In addition, lameness is a condition that is strongly associated with housing and management practices on the farm [17, 25, 68]. As a consequence, farm-level lameness prevalence may reflect a certain type of on farm setting regarding management elements and housing conditions which may translate into additional animal health and welfare related issues. This may further characterise farms, which is the reason we chose this variable as an external predictor in our analysis.

As for the *O. ostertagi* cluster analysis in study region North, BCS appeared to be a relevant factor to discriminate clusters. Body condition scoring is an invaluable tool to assess animal health, production level, and management of dairy cows [69, 70]. A generally lower BCS on farm level may hence indicate impaired health of cows or management shortcomings. Parasitic infections have been demonstrated to be associated with a lower BCS [8, 22, 71] which renders it most plausible that a BCS was identified as an important criterion to differ between parasite–positive and negative farms. Moreover, BCS has been associated with production parameters, which further supports the outcome of this variable separating clusters [72, 73].

Herd size was ranked fourth by the random forest for the *O. ostertagi* cluster analysis in study region North and fifth in the *F. hepatica* cluster analysis in study region South. Herd size may reflect different farm characteristics. For example, large farms may follow more intense managing procedures and feeding regimes. On the other hand, small farms may rather be organically managed or provide pasture access. These characteristics mediated by herd size translate not only into production parameters, but also into parasite status as production levels may differ between small and large herds and the risk of parasite infection also has been associated with herd size [62].

In regions North and East, DIM was identified as an important variable differentiating clusters for the *O. ostertagi* analysis. According to Caffin et al. [74], the selective transport of antibodies to the udder decreases during peak lactation. Furthermore, in late lactation, antibodies are constantly transported to the mammary gland while milk production drops. This may translate into the level of specific antibodies against *O. ostertagi* detected by the ELISA used in the present study [75, 76]. On the other hand, stage of lactation has an effect on production parameters [77, 78], i. e. milk yield, milk fat, milk protein, all input variables of the cluster analyses, which could also explain the relevance of this variable in discriminating clusters.

The random forest of the cluster analysis in study region North revealed the relevance of SCC as a factor differing between clusters. This is plausible given the fact that SCC, being a broadly used indicator for mastitis, has been shown to be correlated with an optical density ratio (ODR) for *O. ostertagi* in milk samples. Charlier et al. [79] stated that acute mastitis lead to an subsequent increase in the *O. ostertagi* ODR values. Apart from that, SCC is associated with other input variables of the cluster analysis such as milk yield [80]. Furthermore, SCC has been associated with lameness [17], the top ranking variable discriminating clusters in study region North. Together, both factors are very common in dairy production and may act as a representation of certain type of farm management and housing conditions.

One striking advantage of the present study is the evaluation of a total number of 606 dairy farms with more than 50,000 cows in three structurally different dairy regions in Germany. It is paramount to be aware of regional variations which can be marked in the dairy sector in Germany [18], which necessitates that epidemiological studies are stratified by regions to obtain valid and reliable results. This was the main rationale for conducting analyses separately for the three study regions. As a wide variety of dairy cow management systems and local characteristics were included within the current study, the results not only are applicable to the entire country but we are also confident that the results from the present work can be extrapolated to other countries and the range of different systems in dairy production. Furthermore, the evaluation of data from all three study regions could provide the possibility to reveal

relevant factors which may not have been uncovered by solely analysing one region or a smaller sample size.

However, the cross-sectional nature of the present study has several limitations inherent to the study design itself [81] which need to be addressed. As potential target variables as well as predictors are simultaneously assessed, a certain degree of bias can enter the data collection at this stage. Even though we cannot entirely rule out the incorporation of bias during the farm visits, we are confident to have minimised the risk of assessment bias by the implementation of a strict study protocol as well as concise standard operating procedures as well as continuing assessment of observers. Secondly, the cross–sectional study design does not allow for the inference of causality among variables. For this purpose, specific study designs ought to be used. One important feature of the present study was the voluntary participation of farmers which may have created a certain level of selection bias by either encouraging proactive farms with improved housing and management conditions or by motivating those farms to partici-pate, which are dealing with specific problems in their management. As outlined previously, farmer characteristics are crucial regarding animal health as well as the intrinsic compliance with external consultation [82–84]. The present study as well as the sampling procedure were based on randomisation in each of the study regions in alignment with region–specific herd size. Although we cannot entirely rule out selection bias in this context, we believe the intro-duction of bias was minimised by the way farm selection was randomised. Additionally, all outcomes are in accordance with the literature, which further corroborates our results.

A bayesian non–parametric bootstrap approach was chosen to calculate bayesian medians. A non-parametric bootstrap can be implemented to estimate a parameter from a set of observa-tions where the assumption about the distribution can be relaxed. Repeated measures for pro-duction parameters (milk yield, milk fat, milk protein, SCC) were present for each of the cows within the data set up to 12 months prior to this study. In order to be able to compare farms, one single value per farm was necessary for the clustering procedure. We furthermore had a large set of data of which we intended to use the entirety of available values for each animal to be subse-quently transferred to farm–level. According to Rubin [85], who introduced the bayesian boot-strap, the most evident advantage of this method is the likelihood statement made about a parameter rather than just a frequency statement. Therefore, the most plausible value reflecting the individual animal in a specific farm setting could be used to transfer the animal-level infor-mation on farm–level during a second round of bootstrapping to obtain farm–level bayesian medians. Since we intended to compare farms, the principal aim was to conserve the highest level of information in regard to the individual animal as well as subsequently in regard to the single farm. Generating a simple median would have entailed a considerable loss of information. A bayesian bootstrap however represented an innovative, promising method to obtain the most plausible and most reliable values reflecting different farms.

We were able to detect farm–level patterns of parameters of milk composition and the BTM *F. hepatica* and *O. ostertagi* antibody status in a large set of farms across three structurally different regions in Germany. Co–infections with *F. hepatica* or *O. ostertagi*, respectively, pas-ture access, organic farming, BCS, farm level lameness prevalence, DIM, SCC, and herd size appeared to discriminate clusters confirmed by the random forest classifier. One striking find-ing of our study is the fact that the cluster analysis detected evident differences between para-site–positive and negative farms without any supervision solely based on the incorporated data. This was further supported by the biological reasoning in regard to other external predic-tors which render the findings even more plausible. K–mode clustering and partitioning around medoids proved to be useful and innovative tools to handle complex data. This approach represents a promising tool for the evaluation of complex settings in a biological or veterinary context.

## Conclusions

Based on the findings of the current study, it is reasonable to infer that considerable and biologically relevant differences exist between farms positive for *F. hepatica* or *O. ostertagi*, respectively and negative farms. An unsupervised cluster analysis was implemented using the partitioning around medoids algorithm. The partitioning around medoids algorithm of the present study confirmed previous evidence and shed further light on the complex systems of associations a between parasitic diseases, milk yield and milk constituents, and management practices. Future efforts may well use the presented approach on a broader scale to gain insights and identify relevant factors in complex disease settings.

## Supporting information

**S1 Table. List of R packages.** A complete list of implemented R packages in alphabetical order.
(PDF)

**S1 Fig. Network structure of interrelationships among variables included in the random forest.** Input variables of cluster analyses and potential associations with external factors.
(TIF)

**S2 Fig. ROC curves of the random forests.**
(TIF)

**S3 Fig. Silhouette plots of each cluster analysis.** Across all cluster analyses, silhouette plots suggested k = 2 clusters to be the most appropriate number of clusters given the underlying data.
(TIF)

**S1 Data.**
(CSV)

**S2 Data.**
(CSV)

**S3 Data.**
(CSV)

**S1 File.**
(PDF)

**S2 File.**
(PDF)

**S3 File.**
(PDF)

**S4 File.**
(PDF)

## Acknowledgments

We would like to thank all participating farmers as well our project colleagues involved in the study (in alphabetical order of their surnames): Friedemann Adler (Department of Biometry, Epidemiology and Information Processing, University of Veterinary Medicine Hannover),

Heidi Arndt (Clinic for Cattle, University of Veterinary Medicine Hannover), Katrin Birnstiel (Clinic for Cattle, University of Veterinary Medicine Hannover), Amely Campe (Department of Biometry, Epidemiology and Information Processing, University of Veterinary Medicine Hannover), Alexander Choucair (Clinic for Ruminants and Swine, Freie Universität Berlin), Phuong Do Duc (Clinic for Cattle, University of Veterinary Medicine Hannover), Antonia Hentzsch (Clinic for Ruminants and Swine, Freie Universität Berlin), Martina Hoedemaker (Clinic for Cattle, University of Veterinary Medicine Hannover), Verena Kaufmann (Clinic for Ruminants and Swine, Freie Universität Berlin), Laura Kellermann (Clinic for Ruminants with Ambulatory and Herd Health Services, Ludwig-Maximilians-Universität Munich), Marcus Klawitter (Clinic for Ruminants and Swine, Freie Universität Berlin), Corinna Lausch (Clinic for Ruminants with Ambulatory and Herd Health Services, Ludwig-Maximilians-Universität Munich), Roswitha Merle (Institute for Veterinary Epidemiology and Biostatistics, Freie Universität Berlin), Moritz Metzner (Clinic for Ruminants with Ambulatory and Herd Health Services, Ludwig-Maximilians-Universität Munich), Kerstin–Elisabeth Müller (Clinic for Ruminants and Swine, Freie Universität Berlin), Philip Paul (Clinic for Ruminants with Ambulatory and Herd Health Services, Ludwig-Maximilians-Universität Munich), Frederike Reichmann (Clinic for Ruminants with Ambulatory and Herd Health Services, Ludwig-Maximilians-Universität Munich), Anne-Sophie Rössler (Clinic for Ruminants with Ambulatory and Herd Health Services, Ludwig-Maximilians-Universität Munich), Dmitrij Sartison (Department of Biometry, Epidemiology and Information Processing, University of Veterinary Medicine Hannover), Alexander Stoll (Clinic for Ruminants with Ambulatory and Herd Health Services, Ludwig-Maximilians-Universität Munich), Annegret Stock (Clinic for Ruminants and Swine, Freie Universität Berlin), Maria Volkmann (Institute for Veterinary Epidemiology and Biostatistics, Freie Universität Berlin), Marina Volland (Clinic for Ruminants and Swine, Freie Universität Berlin), Svenja Woudstra (Clinic for Cattle, University of Veterinary Medicine Hannover), Philip Zuz (Clinic for Ruminants and Swine, Freie Universität Berlin).

## Author Contributions

**Conceptualization:** Andreas W. Oehm, Yury Zablotski.

**Data curation:** Andreas W. Oehm, Katharina Charlotte Jensen.

**Formal analysis:** Andreas W. Oehm, Yury Zablotski.

**Investigation:** Andreas W. Oehm, Andrea Springer, Daniela Jordan, Christina Strube, Gabriela Knubben-Schweizer, Katharina Charlotte Jensen, Yury Zablotski.

**Methodology:** Andreas W. Oehm, Andrea Springer, Daniela Jordan, Christina Strube, Katharina Charlotte Jensen, Yury Zablotski.

**Software:** Andreas W. Oehm.

**Supervision:** Andreas W. Oehm, Christina Strube, Gabriela Knubben-Schweizer, Yury Zablotski.

**Validation:** Andreas W. Oehm, Yury Zablotski.

**Visualization:** Andreas W. Oehm, Yury Zablotski.

**Writing – original draft:** Andreas W. Oehm.

**Writing – review & editing:** Andreas W. Oehm, Andrea Springer, Christina Strube, Gabriela Knubben-Schweizer, Katharina Charlotte Jensen, Yury Zablotski.

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
