## [Decision Letter · Decision Letter 0]

22 Apr 2022

PONE-D-22-04434A machine learning approach using k – mode clustering and random forest classification to model groups of production parameters and bulk tank milk antibody status of two major internal parasites in dairy cowsPLOS ONE

Dear Dr. Oehm,

Thank you for submitting your manuscript to PLOS ONE. After careful consideration, we feel that it has merit but does not fully meet PLOS ONE’s publication criteria as it currently stands. Therefore, we invite you to submit a revised version of the manuscript that addresses the points raised during the review process.

We look forward to receiving your revised manuscript.

Kind regards,

Rebecca Lee Smith, D.V.M., M.S., Ph.D.

Academic Editor

PLOS ONE

Journal Requirements:

“Farm visits and data collection in the context of the underlying cross-sectional study were financially supported by the German Federal Ministry of Food and Agriculture (BMEL) through the Federal Office for Agriculture and Food (BLE) grant number 2814HS008.”

Additional Editor Comments (if provided):

Please be certain to address all reviewer concerns, particularly as to terminology and full description of data.

Reviewers' comments:

Reviewer's Responses to Questions

**Comments to the Author**

1. Is the manuscript technically sound, and do the data support the conclusions?

Reviewer #1: No

Reviewer #2: Yes

2. Has the statistical analysis been performed appropriately and rigorously? 

Reviewer #1: No

Reviewer #2: Yes

3. Have the authors made all data underlying the findings in their manuscript fully available?

Reviewer #1: No

Reviewer #2: Yes

4. Is the manuscript presented in an intelligible fashion and written in standard English?

Reviewer #1: Yes

Reviewer #2: Yes

5. Review Comments to the Author

Reviewer #1: In the title the authors write "k-mode clustering" but in the method section it states "K-medoids" without mentioning "k-mode"(line 207). It should be noted that these terms and the corresponding algorithms are very different.

title - What are these "groups of production parameters and bulk tank milk antibody status"? I don't see any groups of parameters reported. The groups should be of the farms according to the context.

line 67, This sentence is confusing. what is the difference between the terms "cluster" and "group" in this sentence. For clustering analyses, the cluster is the group.

line 191, "Missing values were excluded from" -> Farms with missing values were excluded

Line 192, "Apart from" could mean "except for" or " in addition to". Is antibody status in or not in the model? I think " in addition to" is better here.

line 213 "PAM replaces centroids with medoids"-> PAM replaces means with medoids

line 218 "Classification of Clusters by means of Random Forest". Clustering analyses is for unsupervised classification, I don't see why clusters need to be classified again. Besides, there are only 2 cluster reported (figure 1), how could these be classified again?

Reviewer #2: The manuscript describes a strong and novel work that used k-mode clustering to separate the internal features of the different diary farms and used random forest to analyze features that are important to the cluster separation. The authors used very rigorous statistical approaches in data acquisition, cleaning, aggregation, and analysis. The authors clearly demonstrated the methodology and results, and come to a well-supported conclusion that the antibody status of two major internal parasites in diary cows strongly affects the milk yield and nutrition content. The work thoroughly covered a vast number of farms over major diary production regions in Germany. The conclusion was also supported by relevant studies and field. In addition to the scientific insight that brings direct economical benefits, the manuscript demonstrated the feasibility of using un-supervised methods to perform non-biased analyze on the topic. Therefore, I strongly recommend publication of the work.

6. PLOS authors have the option to publish the peer review history of their article (what does this mean?). If published, this will include your full peer review and any attached files.

Reviewer #1: No

Reviewer #2: **Yes: **Weihao Ge

---

## [Author Response · Author response to Decision Letter 0]

25 Apr 2022

Dear Editor

Dear Reviewer 1

Dear Reviewer 2

We wish to express our gratitude to each of you for taking the time and effort to critically revise our manuscript. We appreciate the overall supportive feedback and feel that the raised points are worth being addressed. Please find below your remarks and our replies. While working on the improvement of this work, we have applied great care to comply with your comments. If there are any further points that need to be modified, we are happy to discuss.

We have modified the naming for the supplementary files both within the main body of the manuscript as well as of the uploaded files. We furthermore have effected some minor modifications within the references section.

“Farm visits and data collection in the context of the underlying cross-sectional study were financially supported by the German Federal Ministry of Food and Agriculture (BMEL) through the Federal Office for Agriculture and Food (BLE) grant number 2814HS008.”

We have added the required information that the funders had no role in study design, data collection and analysis, decision to publish, or preparation of the manuscript. We have also added this to the cover letter.

Reviewer #1: In the title the authors write "k-mode clustering" but in the method section it states "K-medoids" without mentioning "k-mode"(line 207). It should be noted that these terms and the corresponding algorithms are very different.

We have modified the title accordingly

title - What are these "groups of production parameters and bulk tank milk antibody status"? I don't see any groups of parameters reported. The groups should be of the farms according to the context.

Yes, the groups are indeed the farms. in fact, groups of farms were modelled. We have modified the title, to further clarify.

line 67, This sentence is confusing. what is the difference between the terms "cluster" and "group" in this sentence. For clustering analyses, the cluster is the group.

We can understand the confusion in this regard and have modified the sentence.

line 191, "Missing values were excluded from" -> Farms with missing values were excluded

Effected.

Line 192, "Apart from" could mean "except for" or " in addition to". Is antibody status in or not in the model? I think " in addition to" is better here.

Thank you. We have modified the expression in alignment with your remark.

line 213 "PAM replaces centroids with medoids"-> PAM replaces means with medoids

Effected.

line 218 "Classification of Clusters by means of Random Forest". Clustering analyses is for unsupervised classification, I don't see why clusters need to be classified again. Besides, there are only 2 cluster reported (figure 1), how could these be classified again?

The cluster analysis solely grouped the farms together ion the first place. As you said, this is already a sort of classification by sorting similar data point (i.e. farms together). However, after this initial clustering classification (i.e. assignment of farms to one of the two clusters within each study region), we only know that within each study region apparently there are two different groups of farms represented by the clusters. So far, so good. Our intention was to further understand what was behind these two groups of farms meaning how can the farms be characterised or rather how can both clusters be characterised or differentiated based on further variables. Therefore, after the initial sorting of farms into the two clusters, we took external/further variables which initially were not used for cluster creation and which were potentially related to the input variables of the cluster analyses (please refer to the network structure where we provided illustration on this) in order to find out how these external/further factors could help characterising the clusters. This was done using the random forest approach which allowed for the identification of factors that could characterise and differentiate the initially formed clusters and for finding the most important variables reflecting differences among clusters. To avoid confusion, we have replaced the term “classification” with the term “characterisation" in order to render it more clearly. Furthermore, we have added some more information to the first sentence of this paragraph.

Reviewer #2: The manuscript describes a strong and novel work that used k-mode clustering to separate the internal features of the different diary farms and used random forest to analyze features that are important to the cluster separation. The authors used very rigorous statistical approaches in data acquisition, cleaning, aggregation, and analysis. The authors clearly demonstrated the methodology and results, and come to a well-supported conclusion that the antibody status of two major internal parasites in diary cows strongly affects the milk yield and nutrition content. The work thoroughly covered a vast number of farms over major diary production regions in Germany. The conclusion was also supported by relevant studies and field. In addition to the scientific insight that brings direct economical benefits, the manuscript demonstrated the feasibility of using un-supervised methods to perform non-biased analyze on the topic. Therefore, I strongly recommend publication of the work.

We kindly appreciate this very positive and supportive feedback and are happy about the support.

---

## [Decision Letter · Decision Letter 1]

21 Jun 2022

PONE-D-22-04434R1A machine learning approach using partitioning around medoids clustering and random forest classification to model groups of farms in regard to production parameters and bulk tank milk antibody status of two major internal parasites in dairy cowsPLOS ONE

Dear Dr. Oehm,

Thank you for submitting your manuscript to PLOS ONE. After careful consideration, we feel that it has merit but does not fully meet PLOS ONE’s publication criteria as it currently stands. Therefore, we invite you to submit a revised version of the manuscript that addresses the points raised during the review process.

The reviewers have suggested some minor changes to improve the manuscript quality.

We look forward to receiving your revised manuscript.

Kind regards,

Rebecca Lee Smith, D.V.M., M.S., Ph.D.

Academic Editor

PLOS ONE

Journal Requirements:

Additional Editor Comments:

Please consider the recommendations of the reviewer.

Reviewers' comments:

Reviewer's Responses to Questions

**Comments to the Author**

1. If the authors have adequately addressed your comments raised in a previous round of review and you feel that this manuscript is now acceptable for publication, you may indicate that here to bypass the “Comments to the Author” section, enter your conflict of interest statement in the “Confidential to Editor” section, and submit your "Accept" recommendation.

Reviewer #2: All comments have been addressed

Reviewer #3: (No Response)

2. Is the manuscript technically sound, and do the data support the conclusions?

Reviewer #2: Yes

Reviewer #3: Yes

3. Has the statistical analysis been performed appropriately and rigorously? 

Reviewer #2: Yes

Reviewer #3: Yes

4. Have the authors made all data underlying the findings in their manuscript fully available?

Reviewer #2: Yes

Reviewer #3: Yes

5. Is the manuscript presented in an intelligible fashion and written in standard English?

Reviewer #2: Yes

Reviewer #3: Yes

6. Review Comments to the Author

Reviewer #2: The manuscript is strong. The work applied an unsupervised approach that naturally separated the studied farms by positive/negative of common bacterial antibodies. With further feature analysis, the work identified the variables that are associated with infections as well as milk yield. The work is of important direct economical impact and is based on sound statistics. Therefore, I recommend the publication of the work.

Reviewer #3: The results presented in this paper are interesting and of practical importance in identifying biologically relevant differences between farms positive for F. hepatica or O. ostertagi, respectively and negative farms using farm-level bulk tank milk. This important in the veterinary field.

The revisions suggested are minor and involve some additional explanations, plots, and/or statistical procedures.

Minor Revisions and comments:

1. p. 9, line 200: The authors have chosen the Gower’s distance matrix for clustering stating that it is the most common distance matrix for a mix of categorical and continuous values and cite 2 papers (28, 29). It is probably the first distance measure proposed in 1971, but there are currently many more choices. An unsupervised random forest can be used to obtain a proximity matrix that could also be used for clustering. A random forest easily handles mixed types of data. See Conrad and Bailey 2015 PLoS One paper on clustering Cystic Fibrosis patients, for an example. It will probability not make much difference in the clustering results, but the authors should be aware of other measures.

2. p. 15, line 303: For all cluster analyses, the silhouette method selected 2 clusters to be optimal …

Authors should provide an average silhouette plot that shows that 2 clusters are optimal. This is because in the Cluster Analyses Section and the description of Fig 1 (F. hepatica cluster analyses) and Fig 2 (O. ostertagi cluster analyses) there were often a “majority cluster” and a “mixed cluster”. It would be very interesting to know that if k=3 clusters were chosen (assuming that the average silhouette plot showed that 2 or 3 clusters were reasonable choices), if the mixed cluster divided into 2 additional more homogeneous or identifiable clusters.

3. p. 27-28 Section: Classification of Clusters by means of Random Forest

Results of a supervised random forest and variable importance plots are given in

Fig 3 (F. hepatica for North (Fig 3 A) and South (Fig 3 B)) and Fig 4 (O. ostertagi for North (Fig 4 A) and East (Fig 4 B) and South (Fig 4 C)).

Statements are made about what variable are “most important”. This is obtained from the ranking of the variables. There is an rfPermute package that will perform a permutation test and provide estimated permutation p-values for the importance metric of the random forest by permuting the response variable. This would allow the authors to make statements about important variables with a reasonable p-value cut-off. This will strengthen the statements about identifying variables that are most important in classifying or separating the clusters.

4. p. 2: In the Abstract it is stated

“Across all study regions, co-infections with F. hepatica or O. ostertagi, respectively, farming type, and pasture access appeared to be the most important factors discriminating clusters (i.e.39 farms). Furthermore, herd size, BCS, and stage of lactation were relevant criteria distinguishing clusters.” Permutation p-values will allow the statement to be made based using a p-value cut-off.

5. Discussion:

p. 29, line 477: Authors state “Moreover, the current work is probably the first of its kind to implement a k-medoids cluster algorithm and PAM.” Please see and cite:

DJ Conrad, BA Bailey, 2015

Multidimensional clinical phenotyping of an adult cystic fibrosis patient population

PLoS One 10 (3), e0122705

DJ Conrad, J Billings, C Teneback, J Koff, D Rosenbluth, BA Bailey, ..., 2021

Multi-dimensional clinical phenotyping of a national cohort of adult cystic fibrosis patients

Journal of Cystic Fibrosis 20 (1), 91-96

7. PLOS authors have the option to publish the peer review history of their article (what does this mean?). If published, this will include your full peer review and any attached files.

Reviewer #2: **Yes: **Weihao Ge

Reviewer #3: No

---

## [Author Response · Author response to Decision Letter 1]

24 Jun 2022

Reviewer #2: The manuscript is strong. The work applied an unsupervised approach that naturally separated the studied farms by positive/negative of common bacterial antibodies. With further feature analysis, the work identified the variables that are associated with infections as well as milk yield. The work is of important direct economical impact and is based on sound statistics. Therefore, I recommend the publication of the work.

Dear Reviewer 2, we again kindly appreciate your support and positive appraisal of this article.

Reviewer #3: The results presented in this paper are interesting and of practical importance in identifying biologically relevant differences between farms positive for F. hepatica or O. ostertagi, respectively and negative farms using farm-level bulk tank milk. This important in the veterinary field.

Dear Reviewer 3, thank you very much for the positive and constructive feedback to our work. We have the impression that even though they are minor, the points you have raised are important to be addressed and will considerably improve and round up this article. 

The revisions suggested are minor and involve some additional explanations, plots, and/or statistical procedures.

Minor Revisions and comments:

1. p. 9, line 200: The authors have chosen the Gower’s distance matrix for clustering stating that it is the most common distance matrix for a mix of categorical and continuous values and cite 2 papers (28, 29). It is probably the first distance measure proposed in 1971, but there are currently many more choices. An unsupervised random forest can be used to obtain a proximity matrix that could also be used for clustering. A random forest easily handles mixed types of data. See Conrad and Bailey 2015 PLoS One paper on clustering Cystic Fibrosis patients, for an example. It will probability not make much difference in the clustering results, but the authors should be aware of other measures.

Thank you. We have changed the wording of this sentence

2. p. 15, line 303: For all cluster analyses, the silhouette method selected 2 clusters to be optimal …

Authors should provide an average silhouette plot that shows that 2 clusters are optimal. This is because in the Cluster Analyses Section and the description of Fig 1 (F. hepatica cluster analyses) and Fig 2 (O. ostertagi cluster analyses) there were often a “majority cluster” and a “mixed cluster”. It would be very interesting to know that if k=3 clusters were chosen (assuming that the average silhouette plot showed that 2 or 3 clusters were reasonable choices), if the mixed cluster divided into 2 additional more homogeneous or identifiable clusters.

We have provided all silhouette plots for every single one of the cluster analyses. We agree with you that it would have been very interesting -given the constant presence of a «majority cluster» and a «mixed cluster» - to know that if k=3 clusters were chosen (assuming that the average silhouette plot showed that 2 or 3 clusters were reasonable choices), if the mixed cluster divided into 2 additional more homogeneous or identifiable clusters. The silhouette plots yet very unequivocally demonstrated that for the underlying data set a 2-cluster solution appeared to be statistically most appropriate, as the objects match best with their own cluster and poorly with other clusters indicated by a silhouette score of > 0.6. In opposite to that, the silhouette score for a k=3 cluster solution already visibly appeared to be considerably lower. Considering then the subsequent descriptive results from the clustering as well as the random forest results and including biological reasoning in the consideration, the selection of two clusters seems even more plausible and strengthens the work in itself. By providing this piece of information and including all silhouette plots as additional files to this manuscript, we believe that the plausibility, transparency, and traceability of our analyses is ensured.

3. p. 27-28 Section: Classification of Clusters by means of Random Forest

Results of a supervised random forest and variable importance plots are given in

Fig 3 (F. hepatica for North (Fig 3 A) and South (Fig 3 B)) and Fig 4 (O. ostertagi for North (Fig 4 A) and East (Fig 4 B) and South (Fig 4 C)).

Statements are made about what variable are “most important”. This is obtained from the ranking of the variables. There is an rfPermute package that will perform a permutation test and provide estimated permutation p-values for the importance metric of the random forest by permuting the response variable. This would allow the authors to make statements about important variables with a reasonable p-value cut-off. This will strengthen the statements about identifying variables that are most important in classifying or separating the clusters.

We appreciate this very useful suggestion. We have now included a permutation test as suggested using the rfPermute package (also updated in the supplementary file of R packages used). Based on the results, we have made modifications in the results and discussion section (and a short remark in the Materials & Methods Section) in order to present all relevant variables discriminating clusters based on these p values. We are convinced that this course of action has had a very positive impact on the quality of this work and has increased the strength of the conclusions drawn from our analyses. Therefore, we kindly appreciate your comment inciting us to look into this direction.

4. p. 2: In the Abstract it is stated

“Across all study regions, co-infections with F. hepatica or O. ostertagi, respectively, farming type, and pasture access appeared to be the most important factors discriminating clusters (i.e.39 farms). Furthermore, herd size, BCS, and stage of lactation were relevant criteria distinguishing clusters.” Permutation p-values will allow the statement to be made based using a p-value cut-off.

We have now included the estimation of permutation p values within the manuscript to support this statement based on p values. In order to comply with the journal`s guidelines specifically concerning abstract length, which should not exceed 300 words, we have refrained from including these p values into the abstract. 

5. Discussion:

p. 29, line 477: Authors state “Moreover, the current work is probably the first of its kind to implement a k-medoids cluster algorithm and PAM.” Please see and cite:

DJ Conrad, BA Bailey, 2015

Multidimensional clinical phenotyping of an adult cystic fibrosis patient population

PLoS One 10 (3), e0122705

DJ Conrad, J Billings, C Teneback, J Koff, D Rosenbluth, BA Bailey, ..., 2021

Multi-dimensional clinical phenotyping of a national cohort of adult cystic fibrosis patients

Journal of Cystic Fibrosis 20 (1), 91-96

We have rephrased the sentence and included both references

---

## [Editor Report · Decision Letter 2]

30 Jun 2022

A machine learning approach using partitioning around medoids clustering and random forest classification to model groups of farms in regard to production parameters and bulk tank milk antibody status of two major internal parasites in dairy cows

PONE-D-22-04434R2

Dear Dr. Oehm,

We’re pleased to inform you that your manuscript has been judged scientifically suitable for publication and will be formally accepted for publication once it meets all outstanding technical requirements.

Kind regards,

Rebecca Lee Smith, D.V.M., M.S., Ph.D.

Academic Editor

PLOS ONE
---

## [Editor Report · Acceptance letter]

1 Jul 2022

PONE-D-22-04434R2 

A machine learning approach using partitioning around medoids clustering and random forest classification to model groups of farms in regard to production parameters and bulk tank milk antibody status of two major internal parasites in dairy cows 

Dear Dr. Oehm:

I'm pleased to inform you that your manuscript has been deemed suitable for publication in PLOS ONE. Congratulations! Your manuscript is now with our production department. 

Kind regards, 

on behalf of

Dr. Rebecca Lee Smith 

Academic Editor

PLOS ONE